

# The effects of repeated sprints on mid-range and three-point jump shot biomechanics in elite basketball players

Shuairan Li[1,*], Jielin Yang[2,*], Jing Mi[1] and Yue Li[3]

[1] Sports Coaching College, Beijing Sport University, Beijing, China
[2] School of Physical Education, South China University of Technology, Guangzhou, China
[3] School of Physical Education and Sports, Shenzhen University of Information Technology, Shenzhen, China
[*] These authors contributed equally to this work.

## ABSTRACT

**Background**. In basketball, fatigue is a prevalent physiological state that significantly influences shooting mechanics. This study discusses the difference of kinematic parameters of mid-range and long-range basketball players before and after the onset of fatigue, providing a comprehensive analysis of how fatigue affects shooting performance at these distances.

**Methods**. This study involved 12 collegiate male basketball players of at least level I athlete status. Blood lactate levels, subjective exercise fatigue scores, and heart rate variability were used as indices for assessing fatigue. Kinematic data during basketball shooting were collected by inertia motion capture system (Xsens technologies BV, The Netherlands) and a high-speed camera (Sony, Tokyo, Japan). Differences in kinematic parameters between pre- and fatigue conditions during middle and long-range shooting were analyzed using a paired sample T-test.

**Results**. After fatigue, the shooting percentage of middle distance had no significant decrease compared with that before fatigue ($P > 0.05$), while the shooting percentage of long distance had a significant decrease ($P < 0.05$). In the fatigue state, significant decreases in the angular velocity of the wrist ($P < 0.01$) and elbow ($P < 0.05$) were recorded, while an increase was noted in the knee joint's angular velocity ($P < 0.05$). For long-range shots, there was a significant reduction in wrist angular velocity ($P < 0.01$), with marked increases in the angular velocities of the ankle ($P < 0.01$), knee ($P < 0.01$), and elbow ($P < 0.05$).

**Conclusion**. Fatigue has no significant effect on the shooting percentages for mid-range jump shots among elite collegiate athletes; however, it substantially reduced the success rate of long-range jump shots. This decline in performance led to reductions in both the shooting angle and incident angle, as well as the extension of shooting time. Fatigue can compromise the effectiveness of force transfer from the lower to upper extremities. Long-distance shooting requires exceptional lower limb explosive power and precise neuromuscular synchronization.

Corresponding authors
Jing Mi, taishanmijing@126.com
Yue Li, liyuelily2009@163.com

## INTRODUCTION

Basketball is a skill-driven team sport involving speed, acceleration, and change of direction (*Ben Abdelkrim, El Fazaa & El Ati, 2007*). Shooting is the only method of scoring points in basketball, with the jump shot being the most widely used technique in modern basketball games. At the same time, basketball is a high-intensity, intermittent sport. Athletes often need to sprint, slide, jump and stop and start frequently in the game, which requires high endurance, energy metabolism and nervous coordination of athletes (*Abdelkrim, Fazaa & El Ati, 2007*; *Radenković et al., 2022*). High-level basketball competitors can achieve heart rates as high as 90% of peak values, while lactate accumulation in blood samples collected post-competition averages $6.8 \pm 2.8$ mmol/L (*Puente et al., 2017*). With the evolution of game rules, the offensive possession time has been reduced from 30 s to 24 s, resulting in an increased game tempo and significantly raising the physical requirements. The ability to repeatedly perform high-intensity sprints plays a core role in the game and often determines the result (*Girard, Mendez-Villanueva & Bishop, 2011*).

Shooting is the primary method of scoring in basketball. Shooting performance in basketball is primarily determined by three key factors: ball height, release angle, and release velocity. An athlete's standing height, jump height, and the range of motion—particularly the coordination and control of body segments—can all influence the ball release height. *Hudson (1985)* proposed the use of a release height ratio (the ratio of the ball's release height to the athlete's standing height) as a predictor of shooting proficiency and demonstrated that this ratio significantly distinguishes between shooters of different skill levels. Additionally, the release velocity affects the ball's deviation in the anterior-posterior direction. Skilled shooters tend to adopt a release angle that corresponds to the minimum required release velocity, optimizing both accuracy and efficiency (*Okazaki, Rodacki & Satern, 2015*; *França et al., 2021*). It is essential to recognize that the angular velocities of the joints, particularly those in the shoulder, elbow, wrist, and hip, significantly influence these three key factors. As fatigue builds up, the angular velocities of these joints decrease, which can result in alterations to the height, angle, and velocity of the ball release, potentially impairing shooting performance (*Edwards, Wells & Butterly, 2008*; *Heishman et al., 2017*). *Li et al. (2021)* found that under fatigue, not only the angular velocity of the upper limbs during shooting but also during passing and dribbling direction changes decreased more than that of the lower limbs (*Li, Knjaz & Rupčić, 2021*; *Li et al., 2021*). Additionally, there was a notable reduction in the angular velocity of the shoulder, elbow, wrist, and hip joints during passing and changes in dribbling direction ($P < 0.05$). The accumulation of fatigue will have an important impact on athletes' technical performance and overall game outcomes, especially in key shooting techniques. Most studies indicate that fatigue significantly reduces three-point shooting accuracy, whereas its effect on two-point shooting remains debated (*Ardigò et al., 2018*; *Aydemi, 2019*; *Bourdas et al., 2024*; *Brini, Delextrat & Bouassida, 2021*; *Li et al., 2021*; *Marcolin et al., 2018*).

Fatigue plays a crucial role in basketball performance. Given the complexity of fatigue induction, its definition varies across different sports disciplines, and it often requires specialized testing equipment. Previous research has employed artificial intelligence (AI)

techniques to investigate the kinematic characteristics of basketball players performing free throws under pre- and post-fatigue conditions (*Karlik & Hawamdah, 2024*), demonstrating unique patterns and differences in shooting biomechanics caused by fatigue, while underscoring the efficacy of machine learning approaches for examining biomechanical datasets. However, these AI-enhanced studies have focused exclusively on free throws, which represent a relatively static shooting scenario. Jump shooting, the predominant scoring method in competitive basketball, involves significantly more complex biomechanical demands and dynamic movement patterns. Despite its importance, there remains a lack of systematic understanding of how fatigue influences jump shooting mechanics.

Recent advances in motion capture technology have opened new possibilities for detailed biomechanical analysis. Inertial motion capture systems have become increasingly valuable for evaluating movement techniques in high-level athletes, offering more practical and efficient technical support than traditional video analysis methods (*Li, Knjaz & Rupčić, 2021*; *Li et al., 2021*; *Seong et al., 2024*; *Stojanović et al., 2018*; *Wu et al., 2024*). These systems provide precise measurements of joint kinematics without spatial constraints, generating comprehensive datasets suitable for both conventional statistical analysis and future machine learning applications. Therefore, in this study, we used an inertial motion capture system (Xsens Technologies BV, The Netherlands) to assess the impact of physical fatigue on the kinematic parameters of upper and lower limb joints during jump shooting. Specifically, we examined joint angular velocities in the ankles, knees, hips, shoulders, elbows, and wrists, as well as ball release variables. The aim was to quantify changes in shooting technical parameters before and after fatigue, analyze how fatigue influences jump shooting performance, and establish a foundational dataset for future predictive modeling approaches.

## MATERIALS & METHODS

### Participants

Twelve healthy right-handed male basketball players (age: 20.1 ± 0.7 years old; height: 190.1 ± 5.5 cm; body weight: 85.6 ± 3.5 kg; body fat: 11.1 ± 2.6; training experience: 11.8 ± 1.5 years) (Table 1) volunteered to participate in the present study. All participants were professional members of Beijing Sport University Basketball Team and carried out the training workdays as well as competitive weekend games. Therefore, before performing the experiments, they should meet the following special requirements: attended more than 85% of all their teams' trainings; possessed current valid Sports Medicine Certificates; declared that they suffered no illness or injury within three months prior to entering this experiment study design. The research was approved by the Beijing Sport University Institutional Research Commission (Approval number: 2024241H). All participant provided written informed consent in accordance with the Declaration of Helsinki.

### Protocol

Participants were required to refrain from consuming alcohol and caffeine-containing beverages for 24 hours before the test and to fast for 3 hours prior to testing. Each participant

**Table 1  Participants characteristics (mean ± SD).**

| Age (years) | Height (cm) | Weight (kg) | Body Fat (%) | Experience (years) |
|---|---|---|---|---|
| 20.1 ± 0.7 | 190.1 ± 5.5 | 85.6 ± 3.5 | 11.1 ± 2.6 | 11.8 ± 1.5 |

completed all trials during the same time window (14:00–17:00) and under consistent climatic conditions on the test days (temperature: $25° ± 1.5°$), thereby minimizing the impact of circadian variations. All testing procedures were conducted on an indoor basketball court.

The experiment used one group pre-post design. Fatigue was selected as the independent variable, JS% and kinematic parameters were dependent variables. Firstly, athletes wore a Firstbeat heart rate sensor when they performed the Yo-Yo intermittent recovery test level 1. (Yo-Yo IR1; *Castagna et al., 2008*) to measure HRMAX. Then we examined Borg RPE Scale (6-20).

One week later, the athletes carried out official testing. Repeated measurement design includes fatigue condition and non-fatigue condition, and there is at least one-day rest interval before testing. Before starting, we did warm up work: jogging for 5 min, dynamic stretching, jump shot warm-up. After standardized warm-up, subjects began to test, giving them the pass alternately in non-fatigue state, standing 5.80 m (middle distance) and 6.75 m (long distance) away from the top of the arc, respectively. Four jump-shots per each distance. After standardized jump shots practice, induce fatigue by the best test (basketball exercise simulation test). BEST test consists of 24 cycles. Each cycle consists of 30 s of variable-speed running and jumping including walking, jogging, running, sprinting and sliding—actions commonly seen in the game of basketball. The duration of the BEST test is 12 min, without resting between each cycle. It has been shown that it can effectively induce similar external load (*Scanlan et al., 2012*; *Scanlan, Dascombe & Reaburn, 2014*) in the basketball game. We asked players to wear MVN Biomech suit (Xsens Technologies BV, The Netherlands) during the second session. This suit comprises 17 miniature inertial measurement units (nIMUs) strapped to the body. Each nIMU captures the 6-degrees-of-freedom of the body segment to which it is fixed in real-time every second (sampling frequency = 60 Hz).

## Kinematic variables processing

In this study, the calculation of angular velocity typically involves obtaining angular velocity from raw gyroscope data through signal processing algorithms. In practical applications, the calculation of angular velocity may be contaminated by noise, necessitating the use of signal processing techniques such as Kalman filtering to improve the data's stability and accuracy. In Xsens MVN Biomech, these processing steps are typically performed automatically to ensure that the final output angular velocity data is both smooth and reliable. The algorithmic formula is as follows:

$$qnew = (w, x, y, z) \otimes (\cos(\frac{\Delta\theta}{2})), \sin(\frac{wy\Delta t}{2}), \sin(\frac{wz\Delta t}{2}). \tag{1}$$

The extraction of shot parameters was performed using motion analysis techniques, with the Dartfish 2024 motion analysis system employed for video transcription and target

tracking in the analysis module. This allowed for the formation of the ball's flight path, and shot parameters were collected by analyzing indicators such as the ball's displacement and flight time. The angle of shot and the angle of ball entering the basket were derived based on the principles of projectile motion in physics. $y_0$ is the shooting height of the player (from the ground to the hand height); $y_{basket}$ is the height of the basket, usually 3.05 m; $h$ is the height of the player; $g$ is the acceleration of gravity (about 9.81 m/s$^2$); $d$ is the horizontal distance between the player and the basket; $v_0$ is the initial speed of the ball; $\theta_0$ is the angle of the shot. The formula is as follows:

$$y_0 = y_{basket} = h + d \cdot \tan(\theta_0) - \frac{g \times d^2}{2 \cdot (v_0 \cdot \cos\theta_0)^2} \tag{2}$$

$$\theta_{entry} = \arctan(\frac{v_0^2 \sin 2\theta_0}{2gd}). \tag{3}$$

## Statistical analysis

In this study, 192 data of shot kinematics were statistically analyzed by SPSS26.0 (IBM Corp., Armonk, NY, USA). Firstly, calculate the basic description parameters of all measurement variables; secondly, make further treatment and correction on abnormal data according to the specific conditions of normal distribution test (Shapiro–Wilk method). Because shooting scores belong to discontinuous data, use Chi-square method to reduce statistical error; finally, conduct two samples $t$-test. When $p$-value equals or is less than 0.05 in this study, it indicates that results reach significant levels, namely: $P < 0.05$.

# RESULTS

## Exercise protocol

During the first session, maximum heart rate averaged 185.8 ±8.6 bpm, correspond to a total distance of 1,725 m. Following high-intensity exercise, the heart rate (*Rupčić et al., 2020*) of the basketball players increased significantly (non-fatigue condition = 65.60; fatigue condition = 178.25, $P < 0.01$). RMSSD decreased significantly (non-fatigue condition = 47.57; fatigue condition = 4.75, $P < 0.01$), while LF/HF ratios increased significantly (non-fatigue condition = 141.58; fatigue condition = 284.51, $P < 0.01$). Blood lactate concentration (BL) (non-fatigue condition =1.23; fatigue condition = 8.66, $P < 0.01$) and the RPE score (non-fatigue condition = 6.25; fatigue condition = 17.42, $P < 0.01$) also increased significantly. These findings indicate that the fatigue induction protocol in this experiment effectively induced exercise-related fatigue similar to that experienced during competitive play (Table 2).

## Shooting performance

After high-intensity exercise, there was a notable decline in both the shooting accuracy and scoring performance of the athletes during mid- and long-range shooting (Table 3). For mid-range shooting, the shooting percentage (non-fatigue condition = 62.50; fatigue

**Table 2  Changes in internal load indicators before and after fatigue induction.**

| Internal load indicators | Non-fatigue | Fatigue |
|---|---|---|
| HR (times/min) | $65.60 \pm 8.34$ | $178.25 \pm 6.15^{**}$ |
| RMSSD (ms) | $47.57 \pm 14.78$ | $4.75 \pm 1.28^{**}$ |
| LF/HF (%) | $141.58 \pm 86.17$ | $284.51 \pm 101.42^{**}$ |
| BL (mmol/l) | $1.23 \pm 0.41$ | $8.66 \pm 2.36^{**}$ |
| RPE scale (6–20) | $6.25 \pm 0.45$ | $17.42 \pm 0.90^{**}$ |

Notes.

HR, Heart rate; RMSSD, root mean square of successive differences (RMSSD) in R-R intervals; LF/HF, Ratio of low-frequency to high-frequency power in heart rate variability; BL, the players' blood lactate; RPE Scale, the players' rating of perceive exertion.

*Significantly different from non-fatigue, $^*P < 0.05$, $^{**}P < 0.01$.

**Table 3  Comparison of mid-range and long-range shooting accuracy before and after fatigue induction.**

| Distance | Group | Made | Miss | Shooting percentage (%) | Shooting score (points) |
|---|---|---|---|---|---|
| Mid-range shot | Non-fatigue | 30 | 18 | 62.5 | $7.25 \pm 2.34$ |
| | Fatigue | 27 | 21 | 56.3 | $7.04 \pm 2.89$ |
| Long-range shot | Non-fatigue | 22 | 26 | 45.8 | $6.38 \pm 2.82$ |
| | Fatigue | 17 | 31 | $35.4^*$ | $6.00 \pm 2.71$ |

Notes.

*Significantly different from Non-fatigue, $^*P < 0.05$, $^{**}P < 0.01$.

condition = 56.30, $P > 0.05$) and shooting score (non-fatigue condition = 7.25; fatigue condition = 7.04) both decreased, but these changes did not reach statistical significance ($P > 0.05$). However, for long-range shooting, the shooting accuracy fatigue (non-fatigue condition = 45.80; fatigue = 35.40) showed a significant decrease ($P < 0.05$). Although the shooting score (non-fatigue = 6.38; fatigue = 6.00) did not show a significant decline ($P > 0.05$), these results suggest that fatigue has a more pronounced impact on long-range jump shots.

## Shooting kinematic parameters

Kinematic parameters for mid- and long-range shooting, both before and after fatigue, revealed significant differences in the maximum angular velocity of various joints (Table 4, Fig. 1). In mid-range shooting, the maximum angular velocity of the wrist ($P < 0.01$) and elbow ($P = 0.007$) decreased significantly after fatigue, while the maximum angular velocity of the knee increased significantly ($P = 0.046$). In 3-point shooting, the maximum angular velocities of the knee ($P < 0.01$), ankle ($P = 0.002$), and elbow ($P = 0.012$) increased significantly after fatigue, while the wrist's maximum angular velocity significantly decreased ($P < 0.01$).

In terms of shooting parameters, the shooting angle of mid-range jump shots before and after fatigue (non-fatigue = 54.47°; fatigue condition =53.42°) and entry angle (non-fatigue condition = 39.13°; fatigue condition = 37.86°) did not exhibit significant changes ($P > 0.05$). However, the time to execute the mid-range shot significantly increased

**Table 4 Comparison of maximum joint angular velocity for mid-range and long-range shots before and after fatigue induction.**

| Max angular velocity | Mid-range | | P | Long-range shot | | P |
|---|---|---|---|---|---|---|
| | Non-fatigue | Fatigue | | Non-fatigue | Fatigue | |
| Ankle (°/s) | 785.03 ± 178.59 | 803.44 ± 148.76 | 0.115 | 960.44 ± 224.47 | 1,087.04 ± 180.68** | 0.002 |
| Knee (°/s) | 742.73 ± 164.80 | 767.06 ± 117.30* | 0.046 | 881.54 ± 226.26 | 1,074.38 ± 188.66** | 0.000 |
| Hip (°/s) | 387.15 ± 67.58 | 379.73 ± 58.59 | 0.313 | 468.41 ± 59.13 | 484.56 ± 73.08 | 0.083 |
| Wrist (°/s) | 1,733.00 ± 205.56 | 1,518.23 ± 104.43** | 0.000 | 2,474.83 ± 214.98 | 2,202.13.35 ± 212.22** | 0.000 |
| Elbow (°/s) | 1,657.73 ± 400.17 | 1,522.81 ± 115.68** | 0.007 | 1,777.54 ± 281.63 | 1,851.38 ± 251.49* | 0.012 |
| Shoulder (°/s) | 772.52 ± 149.16 | 731.88 ± 140.70 | 0.200 | 884.63 ± 209.26 | 911.29 ± 182.72 | 0.456 |

**Notes.**
*Significantly different from non-fatigue, $*P < 0.05$, $**P < 0.01$.

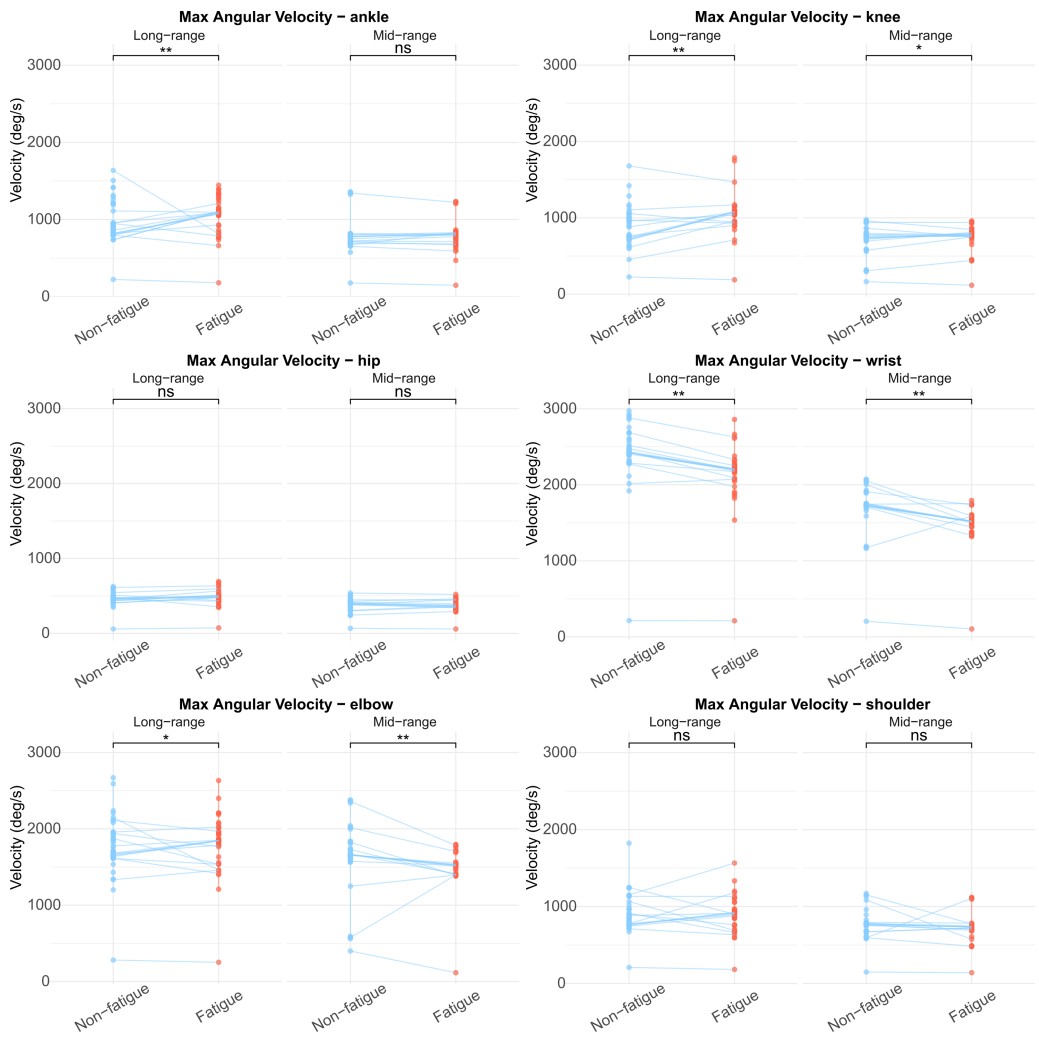

**Figure 1 Comparison of maximum joint angular velocity in medium and long-distance shooting before and after fatigue.**

**Table 5 Changes in shooting parameters for mid-range and long-range shots before and after fatigue induction.**

| Shooting parameters | Non-fatigue | | | Fatigue | | |
|---|---|---|---|---|---|---|
| | Mean ± Std dev. | Max | Min | Mean ± Std dev. | Max | Min |
| Mid-range RA (°) | 54.47 ± 7.52 | 66.70 | 42.10 | 53.42 ± 5.52 | 65.0 | 40.5 |
| Mid-range EA (°) | 39.13 ± 4.46 | 51.00 | 30.60 | 37.86 ± 5.14 | 49.5 | 28.50 |
| Mid-range SS (s) | 0.58 ± 0.10 | 0.79 | 0.40 | 0.65 ± 0.06[*] | 0.87 | 0.50 |
| Long-range RA (°) | 53.50 ± 5.63 | 69.2 | 45.10 | 51.99 ± 6.06[*] | 66.00 | 43.60 |
| Long-range EA (°) | 35.70 ± 5.05 | 44.80 | 25.10 | 34.32 ± 4.43 | 43.50 | 25.40 |
| Long-range SS (°) | 0.65 ± 0.07 | 0.85 | 0.53 | 0.73 ± 0.08[*] | 0.95 | 0.60 |

**Notes.**

RA, release angle; EA, entry angle; SS, shot speed.

*Significantly different from non-fatigue, *$P < 0.05$, **$P < 0.01$.

fatigue (non-fatigue condition = 0.58 s; fatigue condition = 0.65 s) ($P < 0.05$). For long-range shooting, both the release angle (non-fatigue condition = 53.50°; fatigue condition =51.99°) and shot time (non-fatigue condition = 0.65 s; fatigue condition = 0.73 s) significantly increased ($P < 0.05$). In contrast, the entry angle (non-fatigue condition = 35.70°; fatigue condition = 34.32°) did not show significant changes (Table 5, $P > 0.05$).

## DISCUSSION

This study investigated the differential effects of fatigue on kinematic parameters and shooting accuracy in elite collegiate basketball players during mid-range and long-range jump shots, revealing that while mid-range shooting success remained unaffected, fatigue significantly compromised long-range performance, concomitant with marked alterations in joint angular velocities—specifically reductions in upper-limb (wrist and elbow) velocities and increases in lower-limb (ankle and knee) velocities—alongside changes in critical release parameters.

To understand the mechanisms underlying these fatigue-induced changes in shooting performance and kinematics, it is essential to examine the physiological demands and fatigue levels experienced during basketball play. Abdelkrim believe that the average heart rate of athletes in basketball games ranges from 85% to 95% of their maximum heart rate (HRmax), with an average heart rate in the whole game reaching 174 beats/min (*Abdelkrim et al., 2010*; *Girard, Mendez-Villanueva & Bishop, 2011*). Research shows that intense physical exertion can adversely impact several physiological and performance-related metrics in athletes. Specifically, these activities may elevate heart rate and blood lactate levels while simultaneously affecting subjective fatigue perception and athletic performance metrics such as vertical jump height and sprinting speed (*Pernigoni et al., 2024*; *Stojanović et al., 2018*).

The results of the present study show that the average heart rate of elite college basketball players after fatigue intervention reached 178.25 beats per minute (approximately 90% of HRmax), the blood lactate concentration can reach 8.66 mmol/L immediately after exercise, and the average RPE was 17.43. There were significant differences in heart rate variability (*Rupčić et al., 2020*), BL, and RPE between the athletes before and after fatigue

($P < 0.05$), compared to the non-fatigue state. The HRV index showed no recovery signs during the post-exercise period, indicating that the intervention protocol in this study can effectively induce a fatigue state comparable to the game load for basketball players. Therefore, HRV can be considered one of the key indicators for evaluating fatigue in basketball players.

To the best of our knowledge, this is the first study to compare the effects of fatigue on medium- and long-range jump shots. The study findings suggest that fatigue negatively affects shooting accuracy, with slight differences across different shooting distances. *Erculj & Supej (2009)* believes that as fatigue increases, the players' shooting techniques change, especially in terms of the release height of the shoulder and wrist, and the angle of the elbow and upper arm. This automated technique depends on stable neuromuscular patterns, which help maintain consistency in movements and shooting accuracy, even under fatigue (*Ardigò et al., 2018*; *Padulo et al., 2018*).

The accuracy of shooting depends on the consistency and control of the shooting action. The range of motion involved in long-range shooting is larger and includes a motion chain that engages the lower limbs for power generation, trunk rotation, and the upper limbs for shooting. This complex motion chain is more likely to impairment under fatigue, especially with decreased stability in the lower limbs and trunk. This, in turn, reduces the accuracy of shoulder and elbow joint angle control, directly affecting the shooting percentage (*Aydemi, 2019*; *Tsai et al., 2006*). The present study found that long-range shooting is more susceptible to the effects of fatigue, possibly because fatigue compromises joint angle control and movement coordination during the shot. As shooting distance increases there is also the need to increase the impulse from the lower limbs to compensate for the higher distance traveled by the ball (*Miller & Bartlett, 1996*; *Mullineaux & Uhl, 2010*). The significant changes in the variability of coordination—such as the elbow-wrist angle—are also an important mechanism leading to missed shots (*Mullineaux & Uhl, 2010*). Fatigue-induced deterioration in neuromuscular precision can impair the biomechanical kinetic chain responsible for force transmission from proximal to distal segments (*Cao et al., 2022*). Under fatigued conditions, the ability to sustain optimal intermuscular coordination patterns becomes compromised, resulting in diminished postural control and movement consistency. Extended distance shooting necessitates enhanced muscular output, with increasing distances correlating with reduced release angles and elevated ball velocities. To counteract fatigue-related strength deficits, players typically augment lower-extremity propulsion and amplify upper-limb excursion during the shooting motion (*Caseiro et al., 2023*; *Elliott, 1992*; *Elliott & White, 1989*; *Li et al., 2024*).

The kinematic parameters of shooting can reflect the stability and consistency of the shooting action, help to explore the coordination characteristics of the shooting movement, and contribute to a comprehensive understanding of the overall shooting technique. This study found that, under fatigued conditions, the angular velocity of the lower limb joints increased, while the angular velocity of the upper limb wrist joint decreases significantly. This change is consistent with previous findings, suggesting that the angular velocity of distal joints (*e.g.*, elbow and wrist) is transmitted through the angular velocity of proximal joints (*e.g.*, trunk and shoulder) through a dynamic chain (*Erculj & Supej, 2009*; *Li et al., 2024*;

*Li et al., 2021*). This transmission mechanism supports the "proximal-to-distal sequence" movement pattern. In cases where the angular velocity of other joints decreases, the lower limb joints must increase their angular velocity to prevent a reduction in overall jump power from decreasing (*Tsai et al., 2020*). Specifically, as the shooting distance increases, athletes gradually strengthen the angular velocity of their knee, ankle, elbow and shoulder joints to maintain shooting stability and power output. Rupčić et al. believes that fatigue results in a reduction in muscle strength, and that during shooting, the sequence of power transmission may shift from the proximal joints (such as hip and knee joints) to the distal end (such as elbow and wrist) (*Rupčić et al., 2020*). In a fatigued state, athletes are more likely to compensate for the decreased angular velocity of the upper limbs by increasing the angular velocity of the lower limbs. When fatigued, athletes tend to adopt a more upright posture during shooting, which may relate to their adjustment strategy in jump shooting. This adjustment could influence shooting consistency and power distribution, thus altering shooting effectiveness (*Hudson, 1985*; *Uygur et al., 2010*). Uygur et al. found that the shooting effect of the Non-fatigue condition and the fatigue condition did not significantly change before and after fatigue. When players are affected by fatigue, players mainly compensate for the decrease of upper limb angular speed by increasing lower limb angular speed, so as to maintain body stability when shooting. Okazaki believed that long-range shooting demands greater coordination between the upper and lower limbs, and that appropriately increasing the contribution of lower limb power can effectively improve the long-range shooting stability (*Hirashima et al., 2008*; *Okazaki & Rodacki, 2012*; *Okazaki, Rodacki & Satern, 2015*). This study found that after fatigue, with an increase in shooting distance, athletes require more force from their lower limbs to propel and stretch during shooting. Additionally, the aiming point changes, necessitating adjustments in motion control strategies. For instance, athletes may increase angular velocity at the moments of knee and wrist release to optimize the driving force of the shot. At the same time, under fatigue, the maximum angular velocity of the upper limb wrist joint significantly decreases. This may be attributed to muscle fatigue, which impairs neuromuscular fine control and, in turn, affects the coordination and accuracy of shooting movements (*Li et al., 2021*). The synergistic effect of more distal segments may be crucial in jump shots, as the angular velocity of the wrist is primarily derived from the angular velocity of the elbow *via* velocity-dependent torques (*Hirashima et al., 2008*; *Li et al., 2025*).

Shooting is a type of oblique projectile motion. According to the mechanics principles, moderately increasing the projection angle can improve the arc height of the ball and the angle at which it enters the basket, thereby improving the shooting efficacy (*Miller & Bartlett, 1996*). Research shows that as the shooting distance increases, players usually increase the speed of the ball to ensure it reaches the basket. However, the shooting angle and height may vary due to adjustments in the player's technique. In this study, it is found that after fatigue, the time taken for jump shots significantly increases, while both the shooting angle and entry angle decrease. Specifically, the shooting angle for three-point shots is significantly lower after fatigue compared to before. It is suggested that fatigue influences the speed and angle of shooting by changing visual and proprioceptive feedback. Although high-level pitchers have an advantage in force perception and control, fatigue

can still negatively affect overall motor performance. When shooting, various parts of the player's body coordinate to form an efficient movement chain. The adjustment of the angle of incidence needs to consider the coordination of the whole body and the effectiveness of force transmission. *Vencúrik et al. (2021)* believe that in 3-point shots, the entry angle of successful shots is higher than that of missed shots ($P = 0.02$), about 1.6°higher on average. The larger entry angle may be related to the speed of the ball and the force applied when shooting. *Bourdas et al. (2024)* believed that a higher entry angle is usually accompanied by a higher trajectory of the basketball, which can increase the angle at which the ball enters the basket, reducing the probability of the ball striking the rim and, thereby, improving the shooting percentage. *Miller & Bartlett* (*1996*, *1993*) believed that although the theoretically optimal release and entry angles decrease as shooting distance increases, the angles chosen by players in actual games may represent a compromise between considering the actual needs of the game and the maximum fault-tolerant range (*Miller & Bartlett, 1996*; *Miller & Bartlett, 1993*).

Despite its valuable contributions, this study has several limitations. First, all mid-range and three-point shooting tasks were completed in a single testing session, which may have introduced mild recovery effects between trials and potentially reduced the observed impact of fatigue. Second, the shooting tasks were performed in a stationary and isolated manner, lacking the dynamic movement, defensive pressure, and decision-making typically present in real-game situations. Future research should adopt more ecologically valid designs that incorporate continuous movement, reactive shooting, and game-like stimuli to better replicate competitive contexts.

Additionally, the current study focused exclusively on adult, male, elite collegiate basketball players. While this enhances internal validity, it limits the generalizability of the findings. Future studies should investigate whether similar fatigue-induced biomechanical adaptations occur in female athletes, different age groups, and recreational players, as fatigue responses may differ due to variations in physiological and neuromuscular characteristics.

Importantly, this study presents a novel contribution by identifying joint-specific compensatory strategies—specifically, reduced wrist angular velocity and increased lower limb involvement—as potential adaptations to fatigue. These findings suggest that elite athletes may reallocate biomechanical demands to preserve shooting performance under fatigue. Future research should explore these mechanisms further using integrated methods such as electromyography (EMG), artificial intelligence (AI)-based motion analysis, and machine learning models. Such approaches may uncover deeper insights into neuromuscular coordination and movement optimization under fatigue across diverse populations and competition scenarios.

## CONCLUSIONS

Fatigue did not significantly affect the mid-range jump shots success rate of elite collegiate athletes, but it significantly reduced the success rate of long-range jump shots. Fatigue led to a decrease in both the shooting and entry angles, as well as an increase in shot duration for both mid- and long-range shots. In the fatigued state, midrange shooting

increases significantly the angular velocity of the knee joint; long-range shooting presents a significant increase in the angular velocities of the ankle and knee joints. Therefore, it can be speculated that fatigue affects the conversion efficiency between lower limbs and upper limbs during the shooting motion. Additionally, long-range shooting requires greater lower-limb extension velocity, as well as higher muscle coordination. Elite athletes can adjust their neuromuscular systems to maintain shooting accuracy. It is suggested that coaches simulate game conditions as closely as possible in training by incorporating high-intensity shooting drills and small-court game scenarios, which would help players adapt to fatigue and maintain coordinated shooting movements. To sustain scoring effectiveness across the duration of competition, coaching staffs are advised to consider augmenting squad rotation depth. This can be achieved through the strategic allocation of on-court minutes among players in perimeter positions. Future studies should incorporate ecologically valid protocols simulating game conditions with dynamic shooting and defensive pressure. Research should expand to diverse populations across gender, age, and playing positions to enhance generalizability.

## ACKNOWLEDGEMENTS

The authors extend their gratitude to each member of the Men's Basketball Team for their commitment to this work.

### Funding

This work was supported by the Key Laboratory of Sport Training of General Administration of Sport of China, Beijing Sport University, Beijing, China; and the Fundamental Research Funds for the Central Universities (NO. 2024YDXL001). The funders had no role in study design, data collection and analysis, decision to publish, or preparation of the manuscript.

### Grant Disclosures

The following grant information was disclosed by the authors:
Key Laboratory of Sport Training of General Administration of Sport of China, Beijing Sport University, Beijing, China.
Fundamental Research Funds for the Central Universities: 2024YDXL001.

### Competing Interests

The authors declare there are no competing interests.

### Author Contributions

- Shuairan Li conceived and designed the experiments, performed the experiments, analyzed the data, prepared figures and/or tables, authored or reviewed drafts of the article, and approved the final draft.
- Jielin Yang conceived and designed the experiments, prepared figures and/or tables, and approved the final draft.

- Jing Mi conceived and designed the experiments, prepared figures and/or tables, and approved the final draft.
- Yue Li conceived and designed the experiments, performed the experiments, prepared figures and/or tables, and approved the final draft.

## Human Ethics

The following information was supplied relating to ethical approvals (*i.e.*, approving body and any reference numbers):

The research was approved by the Beijing Sport University Institutional Research Commission (Approval number: 2024241H).

## Data Availability

The data are available in the Supplemental Files.

## Supplemental Information

Supplemental information for this article can be found online at http://dx.doi.org/10.7717/peerj.19983#supplemental-information.

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
