# Peer review of "The effects of repeated sprints on mid-range and three-point jump shot biomechanics in elite basketball players"

_PeerJ, doi:10.7717/peerj.19983_

## Round 0.1 · original submission · Major Revisions

Reviewer 1 ·

Basic reporting

Dear authors,

Thank you for the opportunity to review your manuscript entitled “The effects of repeated sprints on mid-range and three-point jump shot biomechanics in elite basketball players”. Please see my comments below.


Introduction
In the Introduction section, I believe it is important to mention the three main variables defining the shooting performance: height, angle, and velocity of ball release. This would provide a better framework for the reader regarding shooting performance. Indeed, angular velocities of the joints will affect the height, angle, and velocity of ball release, influencing the shot output. Please see the references:
Okazaki, V. H., Rodacki, A. L., & Satern, M. N. (2015). A review of the basketball jump shot. Sports biomechanics, 14(2), 190-205.
França, C., Gomes, B. B., Gouveia, É. R., Ihle, A., & Coelho-E-Silva, M. J. (2021). The jump shot performance in youth basketball: a systematic review. International Journal of Environmental Research and Public Health, 18(6), 3283.

In both studies, fatigue is described as an influencer of efficacy, with angular velocities of the shoulder and wrist joint decreasing in fatigued conditions, for instance.
Please identify the study hypothesis based on previous research.

Experimental design

Materials & Methods
I recommend the authors present each shooting protocol using subsections to provide a better reader experience. Overall, it is not clear how many shots were performed in fatigue and non-fatigue conditions. Also, I would recommend providing a graphical presentation of the shooting protocols that have been implemented.
How did you determine peak heart rate? Were participants monitored with a heart rate sensor during the Yo-Yo?
From my understanding, the same participants were included in the fatigue and non-fatigue conditions, correct? If so, I recommend using “condition” instead of “group”.

Validity of the findings

No comments.

Additional comments

Results
Line 170: RPE is mentioned in the results, but its assessment is not mentioned in the Methods section. Please clarify.
Efficacy was considered as a variable, however, in the Methods section this is not described. Please clarify.

Discussion
The first paragraph of the Discussion should describe the study’s objective and main findings.
Line 232-234: As shooting distance increases there is also the need to increase the impulse from the lower limbs to compensate for the higher distance traveled by the ball.
Line 241-242: “The shot velocity increases, and athletes compensate for athletes compensate for the loss of strength...” Please review the sentence.
Instead of “hit rate”, I recommend using “shooting efficacy”.
At the end of the Discussion, please highlight the study’s novelty and suggest future research directions on the topic.

Tables 1 and 3 should include the units for each variable presented.
Table 2 should include the description of each variable in the footnote.

·

Basic reporting

The aim of this study to investigate difference of kinematic parameters of mid-range and long-range
basketball players before and after the onset of fatigue, providing a comprehensive analysis of how
fatigue affect shooting performance at these distances. For this purpose data recorded from 12 collegiate male basketball players.

Experimental design

While the study uses the BEST test and Yo-Yo IR1 protocol to simulate game-like fatigue, the shooting tests are stationary and isolated (i.e., four set jump shots from fixed points). However, real-game shots often involve dynamic movement, defensive pressure, and contextual decision-making. Can the controlled environment truly mirror the complex motor and cognitive demands of live competition?
The authors report decreased wrist angular velocity and increased lower limb angular velocity post-fatigue. This is interpreted as compensation due to impaired upper limb neuromuscular control. However, increased lower limb velocity could reflect a strategic biomechanical adjustment rather than a negative outcome. Could this be an adaptive mechanism that trained athletes use to maintain power transmission under fatigue?

Validity of the findings

The study used 12 elite, right-handed, male, collegiate basketball players from one program. While the internal consistency is strong, this raises questions about external validity. How would these findings hold across:
- Female athletes?
- Younger or older players?
- Recreational or non-elite populations?
- In-game, dynamic shot attempts?
- These questions probe into:
- Ecological validity of testing procedures
- Interpretation of biomechanical adaptations
- Population generalizability

Additional comments

recent literature does show increasing use of machine learning and artificial intelligence (AI) in the biomechanical analysis of basketball shooting, particularly under fatigue. This current manuscript does not sufficiently engage with these emerging methodologies or cite recent influential works. Here are two examples you could suggest the authors examine to improve their literature review:
1. Karlik & Hawamdah (2024)
Title: Artificial Intelligence Enhanced Kinematic Analysis of Basketball Free Throws
2. Rendon-Galvez et al. (2024)
Title: Anthropometric and Electromyographic Characteristics of the Free Throw Shooting Gesture in University Basketball Players.

The authors of the PeerJ manuscript should consider integrating and referencing these newer approaches to:
Highlight methodological gaps between traditional biomechanics and AI-enhanced analytics.
Acknowledge how machine learning can provide predictive insights beyond mean-based statistical comparisons (e.g., T-tests).
Suggest potential for future expansion of their work using ML-based performance modeling.
This would not only modernize their literature review but also position their work within the current research frontier.

---

## Round 0.2 · accepted · Accept

I can confirm that the authors have addressed all of the reviewers’ comments in their revised submission. As the previous reviewers were not re-invited, I have independently assessed the revisions and am satisfied with the amended version of the manuscript. In my opinion, the amended manuscript is now suitable for publication in PeerJ.

Reviewer 1 ·

Basic reporting

Dear authors,

I appreciate your efforts on working on this manuscript. I believe that its overall quality has significantly improved. Thank you for considering my suggestions.

Experimental design

The experimental design is appropriated and the methods section was significantly improved by providing detailed information on data collection protocols and methods implemented.

Validity of the findings

I believe that the findings are robust based on the total number of JS assessed.

Additional comments

Congratulations to the authors on this work.